# Efficient Training for Human Video Generation with Entropy-Guided Prioritized Progressive Learning

## Abstract

Human video generation has advanced rapidly with the development of diffusion models, but the high computational cost and substantial memory consumption associated with training these models on high-resolution, multi-frame data pose significant challenges. In this paper, we propose *Entropy-Guided Prioritized Progressive Learning* (Ent-Prog), an efficient training framework tailored for diffusion models on human video generation. First, we introduce *Conditional Entropy Inflation* (CEI) to assess the importance of different model components on the target conditional generation task, enabling prioritized training of the most critical components. Second, we introduce an *adaptive progressive schedule* that adaptively increases computational complexity during training by measuring the *convergence efficiency*. Ent-Prog reduces both training time and GPU memory consumption while maintaining model performance. Extensive experiments across three datasets, demonstrate the effectiveness of Ent-Prog, achieving up to **2.2×** training speedup and **2.4×** GPU memory reduction without compromising generative performance.

## 1 Introduction

Human video generation (Zhu et al., 2024; Hu, 2024; Xu et al., 2024; Shao et al., 2024), the task of synthesizing realistic video sequences of human actions and appearances, has become essential in applications such as video synthesis, animation, virtual reality, and augmented reality. Recently, diffusion models (Ho et al., 2020) have emerged as a leading framework for video generation due to their ability to produce high-quality, temporally coherent outputs. Video diffusion models (Ho et al., 2022a;b; Khachatryan et al., 2023) generate video sequences by progressively refining noisy inputs, enabling fine-grained control over both spatial and temporal aspects. Condition encoders, such as CLIP visual encoder (Radford et al., 2021), ReferenceNet (Hu, 2024), and ControlNet (Zhang et al., 2023), further enhance the conditional generation ability of diffusion models, allowing the synthesis of high-fidelity videos from diverse reference images and motion controls.

Despite their strengths, training diffusion models for human video generation presents significant computational challenges. The high dimensionality of video data, which involves multiple frames, high resolutions, and intricate temporal dependencies, demands substantial resources. For instance, training a Diffusion Transformer (DiT) (Peebles & Xie, 2023) on videos with a resolution of 512×512 pixels and 20 frames can consume up to 100 GB of VRAM, exceeding the limits of current GPU hardware. Such requirements restrict the scalability to higher resolutions, longer videos, larger model size, and more complex control signals, such as human poses or reference images.

Previous approaches to human video generation often involve adapting a pretrained large diffusion model with fixed configurations, such as pre-defined depth, frame count, and resolution. In conventional deep learning training schemes, all network parameters in the pretrained diffusion model are updated in every training iteration, regardless of their individual contributions to the target task. This static approach can lead to inefficiencies, as it fails to consider that not all parameters or input elements are equally important throughout the training.

To achieve efficient training, a common practice is to minimize the total number of fine-tuning parameters and new parameters. For example, in parameter-efficient tuning methods (Hu et al., 2021),

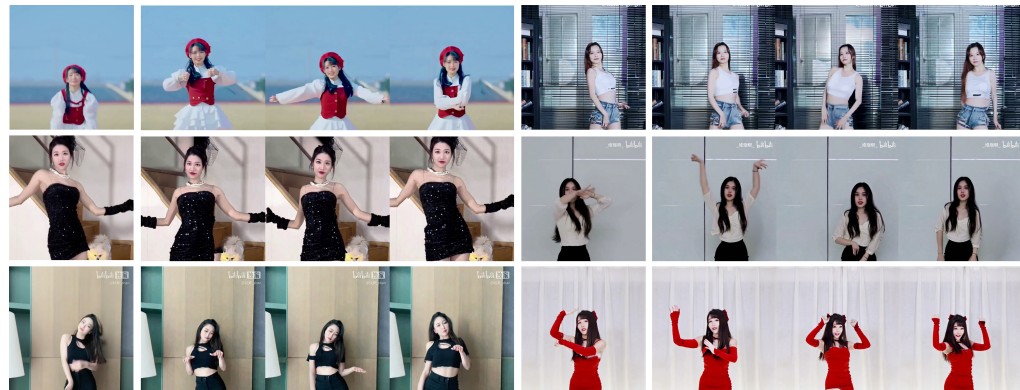

Figure 1: **Human video generation results by Ent-Prog with up to 2.1× training acceleration and 2.4× lower training VRAM usage.** Given a reference image (left image for each clip), we generate consistent and controllable human dance videos after Ent-Prog efficient training.

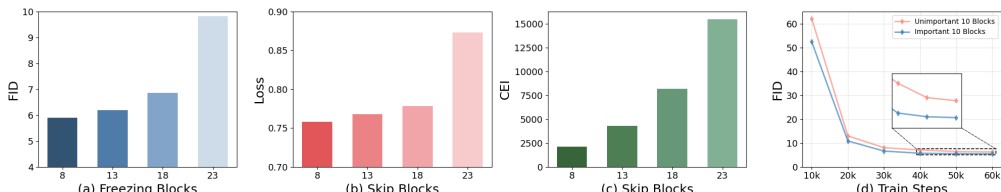

Figure 2: **The impact of freezing or skipping blocks.** (a) illustrates the effect of freezing different numbers of blocks on the model's final convergence performance, showing a clear decline as more blocks are frozen. (b) and (c) present the loss and Conditional Entropy Inflation (CEI) when randomly skipping 8 to 23 blocks, emphasizing the objective of accelerating convergence by selectively skipping blocks with lower interaction (characterized by lower CEI and loss). (d) compares the training dynamics of the most important 10 blocks and the least important 10 blocks, with all other blocks frozen. It is clear that the more influential blocks contribute to faster convergence and better model performance.

total number of fine-tuning parameters is usually set to zero and total number of trainable parameters is set to the minimum via low-rank decomposition. However, when the gap between different task is large, a large capacity of learnable parameters is essential to bypass the gap. Figure 4 (a) illustrates how the number of trainable parameters substantially influences the quality of generated images in pose-guided human image generation.

Effectively identifying which parameters in a pretrained model should be prioritized for training can significantly enhance training efficiency. By randomly skipping 8 to 23 blocks in the pretrained DiT-XL/2 model, the impact on loss and Conditional Entropy Inflation (CEI) in a new task is shown in Figure 4 (b) and (c). As the number of skipped blocks increases, the likelihood of omitting highly interactive blocks rises sharply, leading to network collapse, evidenced by higher loss and increased CEI. Figure 4 (d) presents the training dynamics of the most and least important 10 blocks in the pretrained DiT-XL/2 model on the downstream CUB dataset. Notably, the strongly interactive blocks exhibit significantly faster convergence during training. These observations motivate us to propose a layer importance assessment strategy aimed at optimizing model convergence more effectively.

To address these inefficiencies, we propose *Entropy-Guided Prioritized Progressive Learning* (**Ent-Prog**), an efficient training framework tailored for diffusion models on pose-guided human video generation. *First*, we introduce *Conditional Entropy Inflation* (CEI) to assess the training priority of each network block for the target conditional generation task. CEI measures how much the conditional uncertainty of the predicted noise increases when a block is ablated, providing a task-aware signal of a block's contribution to pose and reference adherence. We define each block's training priority as the Gaussian approximation of CEI, and use these priorities to guide which components should be updated earlier in *prioritized progressive learning*. *Second*, we adopt an *adaptive progressive schedule* to adjust the training load to achieve dynamic balance between performance and efficiency. At the beginning of each stage, we decide how many of the top-priority blocks to un-

freeze by estimating the *convergence efficiency*. A *Nested Diffusion Supernet* provides one-shot estimates across candidate unfreezing sizes and lets us continue the stage with the choice that yields the largest loss decrease per wall time, while reusing supernet parameters. Together, CEI-driven prioritization and adaptive scheduling reduce training time and GPU memory consumption while maintaining pose adherence and generative quality. We demonstrate the effectiveness of Ent-Prog with DiT-based backbone Peebles & Xie (2023) on multiple human video benchmarks.

In summary, our contributions are as follows:

- We propose *Conditional Entropy Inflation* (CEI) to assess the training priority of each network block for the target conditional generation task.

- We further propose the *adaptive progressive schedule* to adjust the training load to achieve dynamic balance between performance and efficiency.

- Ent-Prog accelerates the training process remarkably by up to **2.2×** training speedup and **2.4×** GPU memory reduction without compromising, or even enhancing generative performance across 3 different human video generation datasets.

## 2 ENTROPY-GUIDED ADAPTIVE PROGRESSIVE LEARNING

### 2.1 PRELIMINARIES

**Learning Process of Diffusion Models.** The learning process of diffusion models involves two main phases: the forward diffusion process and the reverse denoising process. The forward diffusion process progressively perturbs a training sample $x_0$ to a noisy version $x_\tau$ by adding Gaussian noise iteratively until timestep $\tau \in [0, 1]$. In the reverse process, the diffusion model progressively denoises the noisy sample to recover the original sample. At each denoising timestep $\tau$, the noise is predicted by a diffusion denoiser network $\hat{\epsilon}(x, \tau)$. The optimization objective of this denoiser network is to minimize the mean square error loss between the predicted noise $\hat{\epsilon}_\omega(x_\tau, \tau)$ and the real added noise $\epsilon$ in the current timestep during forward diffusion.

**Progressive Learning.** Progressive learning incrementally increases a neural network's training load by sequentially expanding its sub-networks used for training, according to a predefined *progressive schedule* $\Psi$. This schedule comprises a series of sub-networks $\psi_k$ with progressively larger sizes over the training epochs $t$. To ensure adequate optimization after each expansion, previous works evenly divide the training process into $|k|$ stages (Yang et al., 2020; Gu et al., 2021; Tan & Le, 2021; Li et al., 2022b). Thus, the progressive schedule is represented as $\Psi = (\psi_k)_{k=1}^{|k|}$, with the final stage corresponding to the complete model. Stages of varying lengths can be achieved by repeating the same sub-network $\psi$ across multiple consecutive stages.

### 2.2 PRIORITIZED PROGRESSIVE LEARNING

Given a *pretrained* diffusion model $\phi(\omega)$ with parameters $\omega$ and residual blocks $\mathcal{B} = \{b_1, \ldots, b_L\}$. When customizing it to a new conditional generation task with training data and corresponding conditions $\mathcal{D} = \{(x_0, c)\}$, our goal is to increase efficiency by *allocating training resources preferentially* to blocks that contribute more to conditional generation on the target task.

**Prioritized progressive learning.** Previous works on progressive learning typically trains from scratch and grows from non-specific sub-networks without prioritizing the training of certain network components. In contrast, when adapting a *pretrained* generative model, different blocks contribute unevenly to conditional generation on the target task. We therefore introduce *Prioritized Progressive Learning* (PPL) to prioritize the training of more important blocks on the target task when progressively unfreezing the network blocks $\mathcal{B}$ in $\phi(\omega)$.

Formally, let $\pi_b \in \mathbb{R}$ denote the score of *training priority* for block $b$. We factorize the progressive schedule into

$$\psi_k = \arg\max_{\psi \subseteq \mathcal{B}} \sum_{b \in \psi} \pi_b \quad \text{s.t.} \quad |\psi| = m_k,$$

where $\psi_k$ is the *unfrozen sub-network*, the set of blocks unfreezed for training at stage $k$, and $m_k$ represents the scheduled number of unfrozen blocks in stage $k$. Intuitively, *high-priority* blocks

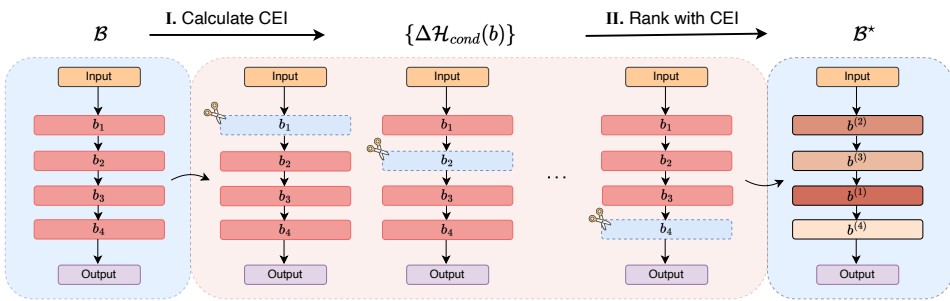

Figure 3: **Illustration of deciding training priority of blocks in diffusion model with *Conditional Entropy Inflation*.** Blocks in deeper colors after ranking indicate higher conditional entropy inflation when the block is skipped, suggesting that the block is more critical for conditional adherence.

start training earlier, while low-priority blocks are deferred. This inproves over previous progressive learning by learning *which* components to emphasize (via $\{\pi_b\}_{b \in \mathcal{B}}$), in addition to *how much* to grow per stage (via $(m_k)_{k=0}^{|k|}$).

We set to answer the remaining questions in the following sections: **1)** how to estimate $\{\pi_b\}$ in a way that is task-aware, and correlates with conditional generation (Section 2.3), and **2)** how to adaptively adjust $m_k$ to achieve a dynamic balance of performance *vs.* efficiency according to the *online performance* of different sub-networks with different number of *unfrozen* blocks (Section 2.4).

## 2.3 Training Priority via Conditional Entropy Inflation

**Conditional entropy in diffusion models.** Conditional diffusion model predicts noise $\hat{\epsilon}$ given noisy latent $\boldsymbol{x}_\tau$, the timestep $\tau$ and the condition $\boldsymbol{c}$. A model that *adheres* to the condition should exhibit a lower uncertainty after observing $\boldsymbol{c}$. This connection is formalized via conditional mutual information:

$$\mathcal{I}(\hat{\epsilon}; \boldsymbol{c} \mid \boldsymbol{x}_\tau, \tau) \,=\, \mathcal{H}(\hat{\epsilon} \mid \boldsymbol{x}_\tau, \tau) \,-\, \mathcal{H}(\hat{\epsilon} \mid \boldsymbol{x}_\tau, \tau, \boldsymbol{c}).$$

When decreasing the conditional entropy $\mathcal{H}(\hat{\epsilon} \mid \boldsymbol{x}_\tau, \tau, \boldsymbol{c})$, the mutual information $\mathcal{I}(\hat{\epsilon}; \boldsymbol{c} \mid \boldsymbol{x}_\tau, \tau)$ increases, i.e., the predictions $\hat{\epsilon}$ carry more information about the condition $\boldsymbol{c}$. In human video generation, it reflects that the generated video adheres better to the reference image and motion condition.

**Conditional Entropy Inflation.** To quantify the importance of each block, we introduce *Conditional Entropy Inflation (CEI)*. CEI measures how much the entropy of the model output increases if a specific block is removed, given the same conditioning state. For block $b$ in the pretrained diffusion model $\phi(\boldsymbol{\omega})$, we compare the conditional entropy of $\hat{\epsilon}$ when the block is skipped versus the original model:

$$\Delta \mathcal{H}_{cond}(b) = \mathcal{H}\big(\hat{\epsilon} \mid \boldsymbol{x}_\tau, \tau, \boldsymbol{c}; \text{skip}(b)\big) \,-\, \mathcal{H}(\hat{\epsilon} \mid \boldsymbol{x}_\tau, \tau, \boldsymbol{c}). \tag{1}$$

A large $\Delta \mathcal{H}_{cond}(b, \tau)$ indicates that skipping block $b$ inflates the uncertainty of the prediction, suggesting that block $b$ is critical for conditional adherence.

Since diffusion models are trained to predict Gaussian noise, we expect the distribution of $\hat{\epsilon}$ to be approximately Gaussian. Under this assumption, we define the score of training priority $\pi(b)$ as the Gaussian approximation of the CEI $\Delta \mathcal{H}_{cond}(b, \tau)$:

$$\pi(b) = \log \frac{\sigma_{\text{skip}(b)}(\hat{\epsilon})}{\sigma(\hat{\epsilon})}, \tag{2}$$

where $\sigma(\hat{\epsilon})$ is the standard deviation of $\hat{\epsilon}$ with the full model, and $\sigma_{\text{skip}(b)}(\hat{\epsilon})$ is the corresponding value when block $b$ is disabled. In practice, we randomly sample around $1,000$ $\tau$ and $(\boldsymbol{x}_t, \boldsymbol{c})$ pairs to calculate $\pi(b)$. Blocks $\mathcal{B}$ are then ranked by $\pi(b)$, with higher-ranked blocks unfrozen earlier, which ensures that training resources are allocated more to components that contribute more to reducing conditional uncertainty.

Figure 4: **Illustration of adaptive progressive schedule.** We search within the defined space to identify sub-networks with optimal convergence efficiency. At the start of each progressive learning stage, we first train a one-shot supernet, and evaluate the convergence efficiency of each unfreezing choice. The sub-network with optimal convergence efficiency is then selected, inheriting parameters from the supernet and continuing training in the next phase.

## 2.4 ADAPTIVE PROGRESSIVE SCHEDULE VIA CONVERGENCE EFFICIENCY

Given the entropy-guided priorities $\{\pi_b\}$, we now determine, at the *beginning of each stage $k$*, how many blocks to unfreeze, $m_k$. Our objective is to *adaptively* set the training load so that each stage uses the unfreezing choice with optimal convergence efficiency on the target objective.

**Nested Diffusion Supernet.** To score the convergence efficiency of each candidate $m$, we employ a *Nested Diffusion Supernet* $\Phi(\widehat{\omega})$ that nests parameters for all unfreezing choices in $\mathcal{M}_k$ under a shared weight space $\widehat{\omega}$. Let $\mathcal{B}^{\star} = (b^{(1)}, \ldots, b^{(L)})$ be the blocks ranked by priority $\pi_b$ (high to low). For a candidate count $m \in \mathcal{M}_k$, we define the unfreezing set $\mathcal{B}_{train}(m) = \{b^{(1)}, \ldots, b^{(m)}\}$. At the start of stage $k$, we train $\Phi$ for one epoch by uniformly sampling the candidate unfreezing choice. At each step, we *randomly sample* an $m \in \mathcal{M}_k$, unfreeze $\mathcal{B}_{train}(m)$, and update only those parameters. During supernet training, the forward pass always uses the full network, while gradients are applied only to blocks in $\mathcal{B}_{train}(m)$, allocating the training resources to prioritized blocks.

**Measuring convergence efficiency.** To take real-world efficiency into consideration, we record the elapsed wall time $\{T_m^{(s)}\}_{s=1}^{S}$ for each training step $s$ of each candidate $m$. After each step of supernet training, we evaluate each $m$ on a fixed small hold-out set $\mathcal{D}_{\text{eval}}$ (part of the training set that are not used in supernet updates) with fixed denoising timesteps $\tau$ and fixed noises, yielding a loss trace $\{\ell_m^{(s)}\}_{s=1}^{S}$ collected at the total $S$ steps during the supernet epoch. We define the *convergence efficiency* of $m$ as the average loss decrease per wall time:

$$\text{CE}(m) \;=\; -\frac{\sum_{s=2}^{S} \left(\ell_m^{(s)} - \ell_m^{(s-1)}\right)}{\sum_{s=2}^{S} \left(T_m^{(s)}\right)}.$$

A larger $\text{CE}(m)$ indicates more efficient convergence. $\text{CE}(m)$ can be seen as an estimation of the *time derivative of loss*. As all candidates share the same initialization at the beginning of the stage, the short-term loss decrease is well-approximated by a first-order Taylor step with $\text{CE}(m)$.

After the one-shot supernet epoch, we select the optimal $m_k^{\star}$ that achieves the highest $\text{CE}(m)$ and *resume* training for the remainder of stage $k$ using the corresponding unfrozen set with the top-$m_k^{\star}$ blocks with highest priority $\pi_b$, *reusing* the supernet parameters $\widehat{\omega}$. This results in an *adaptive progressive schedule* that gradually unfreezes the most important blocks with block number that achieves optimal convergence efficiency.

## 3 EXPERIMENTS

### 3.1 IMPLEMENTATION DETAILS

**Data Preprocessing** We evaluate Ent-Prog on a dataset of approximately 1,000 real-world dance videos collected from Bilibili, following the preprocessing protocol of Open-Sora (Zheng et al., 2024). From these, we extract 4,086 clips for training and reserve 10 full videos for testing. Human pose sequences are estimated using Multi-HMR (Baradel et al., 2024). All experiments are conducted on four A800 GPUs with a unified preprocessing workflow, where frames are cropped according to bounding boxes detected by Yolov7 (Wang et al., 2023a) and resized to 768×768.

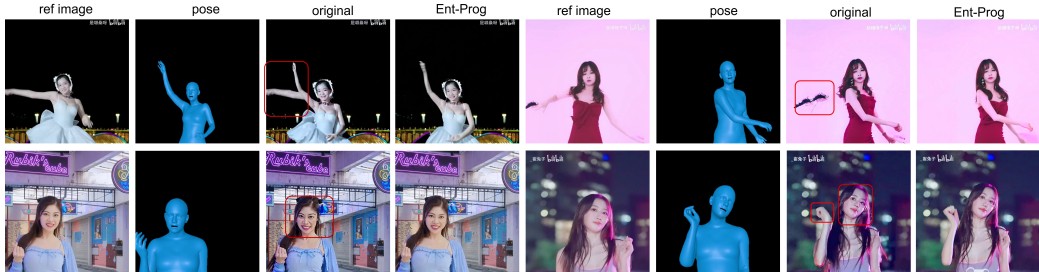

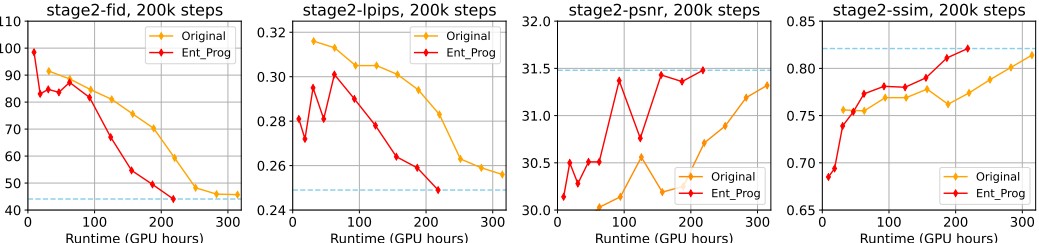

Figure 5: **Qualitative comparison of two training methods on *Bilibili* dataset.** The red boxes indicate the defects of the generated images. Ent-Prog surpasses full training in terms of visual coherence and realism, and it also excels in restoring fine-grained details such as facial expressions. The first row highlights the unreasonable artifacts in the results generated by the full training method, which are absent in Ent-Prog. The second row marks the shortcomings of the full training method in restoring facial features.

Figure 6: **Comparison of evaluation metrics in training progress.** Models are evaluated every 10 epochs in the second stage. Ent-Prog speeds up the training process by $1.45\times$ without compromising performance.

**Training Strategy.** To enhance model transfer learning, we design a multi-stage training strategy tailored to different objectives. The training procedure consists of three stages: **1)** a subject-driven generation stage, where the model is trained for 50k steps with a batch size of 32 to learn subject-driven image generation from a reference image; **2)** a pose-guided generation stage, where the model is trained to generate video frames from both the reference image and control poses, training for 200k steps with a batch size of 8; **3)** a video generation stage, where we only train all the temporal layers on 10-frame sequences resized to 512×512 for 200k steps with a batch size of 4. Across all stages, the learning rate is fixed at 1e-5. The architecture of our baseline model is in Appendix A.

**Inference Setting.** During inference, we use the IDDPM sampler (Nichol & Dhariwal, 2021) with 100 denoising steps and set the classifier-free guidance scale to 4.0. To further validate generalization, we evaluate Ent-Prog on the UBC Fashion Video Dataset and the TikTok Dataset, following established evaluation protocols from prior work (Wang et al., 2023b; Xu et al., 2024; Hu, 2024; Zhu et al., 2024).

**Evaluation Metric.** To validate the effectiveness of our method, we conducted evaluations on the Bilibili dataset as well as two specific benchmarks: human dance generation and fashion video synthesis. For single-frame quality evaluation, we followed the established evaluation metrics used in previous studies, including L1 error, Structural Similarity Index (SSIM; Wang et al. (2004)), Peak Signal-to-Noise Ratio (PSNR; Hore & Ziou (2010)), and Learned Perceptual Image Patch Similarity (LPIPS; Zhang et al. (2018)). Additionally, we used Video-level FID (FID-VID; Balaji et al. (2019)) and Fréchet Video Distance (FVD; Unterthiner et al. (2018)) metrics to evaluate video-level performance.

**Datasets.** We evaluate Ent-Prog on three datasets: Bilibili dataset, TikTok dataset (Jafarian & Park, 2021), and UBC-Fashion dataset (Zablotskaia et al., 2019). These datasets exhibit substantial differences in content, enabling a comprehensive assessment of our method across diverse scenarios. Our evaluation focuses on two tasks: human video generation and image conditioned generation, providing a comprehensive assessment of the model's capabilities in both image and video generation. For image generation, we report results using the model trained in the second stage, while for video generation, we use the model obtained after the third stage of training.

Table 1: **Results comparison of efficient training on *Bilibili* dataset.** The reported training time speedup corresponds to the wall time of the training process.

| Training scheme | L1 ↓ | SSIM ↑ | PSNR ↑ | LPIPS ↓ | FID-VID ↓ | FVD ↓ | Memory (GB) | Speedup |
|---|---|---|---|---|---|---|---|---|
| *100k steps* | | | | | | | | |
| Original | **1.31e-05** | **0.885** | 33.00 | **0.129** | 16.50 | 168.17 | 72 | - |
| Ent-Prog | 1.43e-05 | 0.884 | **33.26** | 0.132 | **15.52** | **120.35** | **44** | **1.52×** |
| *200k steps* | | | | | | | | |
| Original | 1.31e-05 | 0.886 | 33.93 | 0.128 | **15.01** | 132.06 | 72 | - |
| Ent-Prog | **1.26e-05** | **0.892** | **34.41** | **0.121** | 14.90 | **119.77** | **53** | **2.17×** |

Table 2: **Results comparison of human dance image generation** in the first two stages of training on *Bilibili* dataset.

| Training Scheme | L1 ↓ | SSIM ↑ | PSNR ↑ | LPIPS ↓ | FID ↓ | Memory | Speedup |
|---|---|---|---|---|---|---|---|
| *Bilibili stage 1* | | | | | | | |
| Original | 7.72e-06 | 0.952 | 31.48 | 0.068 | 20.93 | 56 | - |
| Ent-Prog | **6.74e-06** | **0.953** | **32.08** | **0.066** | 20.93 | **31** | **2.07×** |
| *Bilibili stage 2* | | | | | | | |
| Original | **2.44e-05** | 0.814 | 31.32 | 0.256 | 45.71 | 77 | - |
| Ent-Prog | 2.61e-05 | **0.821** | **31.48** | **0.249** | **44.09** | **49** | **1.45×** |

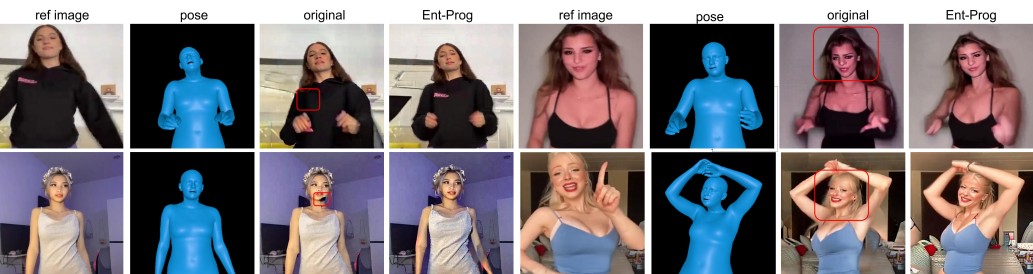

Figure 7: **Qualitative comparison of two training methods on *TikTok* dataset.** The red boxes indicate the defects of the generated videos. With the original training method, the top-left example shows that the logo on the garment is missing, while other examples show distortions in facial details. In contrast, Ent-Prog accurately restores the detailed information from the reference image.

## 3.2 HUMAN DANCE VIDEO GENERATION ON BILIBILI DATASET

Bilibili dataset contains numerous high-resolution single-person dance videos with large motion amplitudes and substantial diversity in scenes and clothing, making it well-suited for evaluating model's generalization ability.

**Human dance video generation.** As shown in Table 1, Ent-Prog surpassed full training in all single-frame quality metrics and video fidelity for human dance video generation, achieving a **2.17×** speedup (**6** vs. **13** days) and reducing GPU memory usage to **45.1%**.

**Human dance image generation.** As shown in Table 2, Ent-Prog consistently outperforms full training across all metrics for human image generation, while achieving **2.07×** training speedup and 46.6% reduction in GPU memory usage during the first training stage, as well as 1.45× training speedup and 36.4% memory reduction during the second training stage. Furthermore, Figure 6 illustrates the changes of four image quality metrics during the second-stage training on BiliBili dataset using Ent-Prog and the full training method. It is evident that, given the same training time, Ent-Prog consistently outperforms full training across all metrics.

**Qualitative comparison.** Figure 5 presents the qualitative comparison results, where Ent-Prog demonstrates superior *adherence to motion guidance* and better preservation of full-body details, outperforming full training in both facial detail and overall fidelity.

## 3.3 HUMAN DANCE VIDEO GENERATION ON TIKTOK DATASET

TikTok dataset contains hundreds of casual, non-professional videos of multiple motions with simple scenes, plain clothing, and gentle movements. As a public benchmark, it has been widely used for evaluation video generation tasks, proving strong credibility.

**Human dance video generation.** Table 3 presents quantification comparison between Ent-Prog and full training on TikTok dataset for video generation. Ent-Prog outperforms full training across all metrics, with a notable **46.1%** relative improvements in FVD. In terms of efficiency, Ent-Prog achieves **1.52×** training speedup and reduces memory usage to **61.1%** of full training.

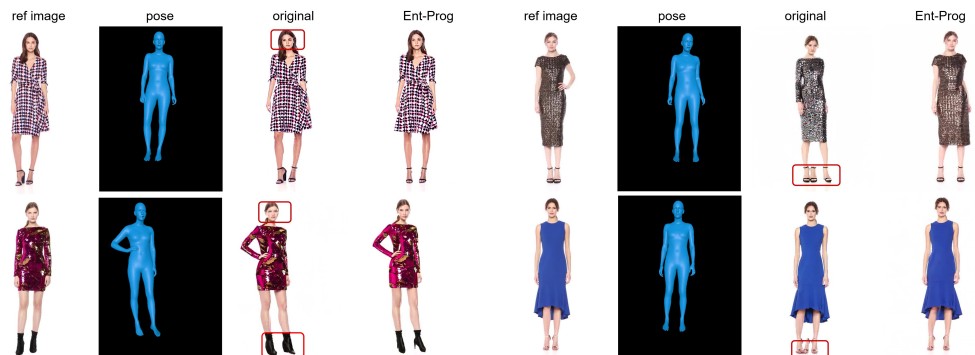

Figure 8: **Qualitative comparison of two training methods on *UBC-Fashion* dataset.** Red boxes highlight artifacts in the generated videos. Ent-Prog accurately and coherently reconstructs the person from the reference image, while the full training method introduces unrealistic flaws, such as the presence of three feet in the two examples on the right side.

Table 3: **Results on human video generation.** Video quality comparison on the Tiktok and UBC dataset.

| Training scheme | SSIM ↑ | PSNR ↑ | LPIPS ↓ | FVD ↓ | Memory (GB) | Speedup |
|---|---|---|---|---|---|---|
| *Tiktok* | | | | | | |
| Original | 0.747 | 29.53 | 0.316 | 385.64 | 72 | - |
| Ent-Prog | **0.790** | **30.77** | **0.268** | **264.03** | **44** | **1.52×** |
| *UBC* | | | | | | |
| Original | 0.906 | 36.44 | 0.069 | 79.94 | 46 | - |
| Ent-Prog | 0.906 | **36.45** | **0.068** | **79.76** | **29** | **1.69×** |

Table 4: **Results on pose-guided image generation.** Comparison of the results between Ent-Prog and Full training in the first two stages of image-conditioned modeling.

| Training Scheme | L1 ↓ | SSIM ↑ | PSNR ↑ | LPIPS ↓ | FID ↓ | Memory | Speedup |
|---|---|---|---|---|---|---|---|
| *Tiktok* | | | | | | | |
| Original | 5.78e-05 | 0.727 | 28.45 | 0.339 | 61.29 | 77 | - |
| Ent-Prog | **5.16e-05** | **0.735** | **28.86** | **0.330** | **59.18** | **32** | **1.45×** |
| *UBC* | | | | | | | |
| Original | 1.46e-05 | **0.887** | 36.17 | 0.082 | 12.13 | 50 | - |
| Ent-Prog | **1.33e-05** | 0.886 | **36.52** | **0.079** | **11.84** | **34** | **1.45×** |

**Human dance image generation.** For image generation, Ent-Prog maintains strong performance compared to full training, while providing **1.45×** training speedup and **58.4%** reduction in GPU memory usage. The quantitative results are summarized in Table 4.

**Qualitative comparison.** Figure 7 presents qualitative comparisons. Ent-Prog consistently recovers *finer details* and better preserves *subject identity*, with noticeably improved facial expression reconstruction. It is noteworthy that in the first example, Ent-Prog better keeps garment details, which the standard fine-tuning baseline omits.

### 3.4 HUMAN FASHION VIDEO GENERATION ON UBC-FASHION DATASET

UBC-Fashion dataset comprises numerous fashion model showcase videos, characterized by highly stylized clothing, and serves as a widely used benchmark in video generation task.

**Human fashion video generation.** Table 3 present the quantitative comparison between Ent-Prog and full training on video generation, where Ent-Prog consistently outperforms full training across all metrics. For acceleration, Ent-Prog achieves **1.69×** training speedup and lowers GPU memory usage to **63.0%** for video generation compared to full training.

**Human fashion image generation.** For human fashion image generation, Ent-Prog consistently maintains high performance, surpassing full training across all metrics, while achieving a **45.4%** acceleration in training and a **32%** reduction in memory usage. The quantitative results are summarized in Table 4.

**Qualitative comparison.** Qualitative examples are shown in Figure 8. It is evident that Ent-Prog performs better in maintaining clothing textures and ensuring motion coordination in fashion videos.

### 3.5 ABLATION STUDY

To demonstrate the effectiveness of the two components of our Ent-Prog, we investigate two alternative designs: **1)** removing the CEI training priority, and **2)** removing the adaptive progressive

schedule and using a linear schedule instead. These variations are compared to our proposed Ent-Prog. We test the video generation capability on the TikTok dataset.

**Effect of CEI.** As shown in Table 5, when removing the CEI training priority, the training scheme fails to find the optimal network configuration for accelerating model convergence, leading to inferior performance. The FID-VID deteriorates sharply *from 32.15 to 37.43*.

**Effect of adaptive progressive schedule.** We further validate the effectiveness of adaptive progressive schedule through ablation experiments. As shown in Table 5, the absence of adaptive progressive schedule leads to a decline in both single-frame and video-level metrics. While removing adaptive progressive schedule can speed up training, it leads to performance degradation due to suboptimal configurations during training.

Table 5: **Ablation on *adaptive progressive schedule* and *CEI*.** We compare video generation results on the TikTok dataset.

| Training scheme | L1 ↓ | SSIM ↑ | PSNR ↑ | LPIPS ↓ | FID-VID ↓ | FVD ↓ | Memory (GB) | Speedup |
|---|---|---|---|---|---|---|---|---|
| Original | 4.64e-05 | 0.747 | 29.53 | 0.316 | 32.85 | 385.64 | 72 | - |
| w/o Adaptive | 3.78e-05 | 0.788 | 30.04 | 0.272 | 44.31 | 382.41 | 28 | 2.11× |
| w/o CEI | 3.79e-05 | 0.789 | 30.51 | 0.270 | 37.43 | 285.94 | 30 | 1.95× |
| Ent-Prog | **3.78e-05** | **0.790** | **30.77** | **0.268** | **32.15** | **264.03** | 44 | 1.52× |

## 4 RELATED WORK

**Diffusion Models for Human Video Generation.** Video generation (Ho et al., 2022a; Khachatryan et al., 2023) builds on image generation by adapting architectures and attention mechanisms (Ho et al., 2022b). A key branch, human video generation (Hu, 2024; Xu et al., 2024; Shao et al., 2024), focuses on producing temporally coherent videos of humans from the given character images and motion sequences. Despite progress, these models remain limited by high computational and memory demands.

**Progressive Learning.** Deep learning models excel across many tasks but incur high training costs, motivating research into efficient methods. Progressive learning (Bengio et al., 2006) has gained attention for its ability to accelerates training without performance loss. Auto-Prog (Li et al., 2022a) improves training efficiency by dynamically adjusting growth schedules, while (Li et al., 2024) extends progressive learning to fine-tuning with progressive unfreezing for diffusion models. Our research further proposes a Entropy-Guided Progressive Learning training framework which tailored for diffusion models.

**Efficient training for diffusion models.** Diffusion models (Ho et al., 2020) achieve strong performance but suffer from high training costs. Efficient fine-tuning methods for diffusion models have been extensively studied. Few-shot methods like DreamBooth (Ruiz et al., 2023) adapt models with limited data, while other works (Rombach et al., 2022; Bao et al., 2022a;b; Lu et al., 2022a;b) target efficiency through sampling strategies. Recent studies (Zhang et al., 2024; Wang et al., 2024) propose multi-stage and patch-level frameworks to cut training costs, though these are often hand-designed and less transferable. In contrast, our method Ent-Prog automates training acceleration, reduces memory use, and generalizes across architectures.

## 5 CONCLUSION

We presented *Entropy-Guided Prioritized Progressive Learning* (Ent-Prog), an efficient training framework for pose-guided human video diffusion. Ent-Prog combines **1)** *Conditional Entropy Inflation* (CEI) to prioritize blocks that most improve pose adherence and **2)** an *adaptive progressive schedule* via a *Nested Diffusion Supernet* to select, at each stage, how many top-priority blocks to unfreeze based on short-horizon convergence efficiency. This concentrates updates where they matter most while keeping full-capacity forward passes, yielding up to **2.2×** training speedup and **2.4×** GPU memory reduction across three datasets, without degrading visual quality or pose alignment.

**Limitation:** Efficient training may result in a proliferation of models with harmful biases or intended uses.

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

## A  MODEL ARCHITECTURE FOR HUMAN VIDEO GENERATION

We propose a strong baseline model for human video generation based on diffusion transformer.

**Transformer-based Denoiser with Temporal Attention.** Our baseline model start with a Diffusion Transformer (DiT) designed for image generation. DiT is a latent diffusion model with VAEs to encode the higher level input to lower dimension latent feature. A transformer-based diffusion model learns the distribution of the latent features. We add a temporal attention layer to each block of the diffusion transformer to model the temporal relationship across frames.

**Siamese ReferenceNet.** ReferenceNet is widely used in controlled video generation as it can capture the detailed appearance of reference image. However, previous works train referencenet as a separate network with its own set of parameters. The optimization of the new set of parameters during training is computationally costly. To reduce GPU memory usage and speedup training, we propose Siamese ReferenceNet, which share exactly the same architecture and parameter of the denoiser network. The Siamese ReferenceNet is connected to the main denoiser network through cross-attention.

**Efficient DiT ControlNet.** We use an efficient transformer-based ControlNet for pose condition. We use the PixArt with only 1 transformer layer to minimize the memory consumption.

## B  ADDITIONAL RESULTS

**Effect of Siamese ReferenceNet.** Given the superior performance of double network structures in generative tasks, we adopt a similar approach inspired by models like Animate Anyone (Hu, 2024) and Champ (Zhu et al., 2024). Our architecture includes an additional DiT-based reference network alongside 1-layer ControlNet blocks. To evaluate the effectiveness of image-conditioned generation, we conduct experiments on the Bilibili dataset. The results, as presented in Table 6, indicate that our original model outperforms the double network structure(Reference Network, RN), achieving competitive results with a simpler architecture and lower memory and computational costs. Notably, our proposed Ent-Prog model achieves the best performance across all metrics, demonstrating enhanced training speed and improved generation quality.

These results qualitatively illustrates the effect of different components on efficient model training. Ent-Prog excels at preserving subject details and accurately synthesizing single-frame images, particularly in retaining fine body details and generating correct images.

Table 6: Comparison with different model structure.

| Training scheme | L1 ↓ | SSIM ↑ | PSNR ↑ | LPIPS ↓ | FID ↓ |
|---|---|---|---|---|---|
| w/o RN | 4.15e-5 | 0.745 | 26.32 | 0.330 | 176.02 |
| Original | 3.11e-5 | 0.718 | 28.42 | 0.279 | 115.42 |
| Ent-Prog | **2.75e-05** | **0.783** | **28.56** | **0.279** | **95.98** |

