# OpenReview forum: "Efficient Training for Human Video Generation with Entropy-Guided Prioritized Progressive Learning"
_ICLR.cc/2026/Conference — ICLR 2026 Conference Withdrawn Submission_

### Official Review · Reviewer_irRC · 2025-10-25

**Soundness:** 2
**Presentation:** 2
**Contribution:** 2
**Rating:** 2
**Confidence:** 5

**Summary:**

This paper proposes Entropy-guide prioritized progressive learning (Ent-Prog) for efficient pose-guided human video generation with diffusion. The idea is to identify the importance of each block in the model, then adopt a progressive training scheme to train from these blocks. A metric call conditional entropy inflation (CEI) is proposed to score the importance of blocks, which is defined as the entropy difference of skip the block or not. Then, according to CEI, prioritized progressive learning is adopted, which starts with the top highest priority blocks unfrozen, then progressively unfreeze more blocks in CEI order. Experiments on several datasets are conducted to shown the effectiveness of the method.

**Strengths:**

1. The method can largely reduce the training cost and memory without quality degradation.

2. The proposed CEI metric can measure the importance of each layer block.

**Weaknesses:**

1. While the CEI is defined as difference of entropy, in practice it is implemented by calculating the ratio of standard deviation in a statistical way, where about 1k samples are sampled. This seems to be inaccurate and the samples are too few. In addition, the author should compare the proposed CEI with other model pruning or layer scoring methods.

2. Another major concern is in experiments. There is no comparison to other competitive methods, like animate anyone, champ, stable animator, etc. Also lack the comparison to other model pruning methods comparing to CEI.

3. Some results in experiments are quite weird. In Table 2, the LPIPS and FID in stage 2 are higher than in stage 1, indicating the results were getting much worse in stage 2. Why that happens?

4. In all experiments, the results of full training are worse than the proposed progressive training. It seems to be a little strange, as I think the proposed method only contribute to the training efficiency, not the performance.

5. As a generation work, there are no videos in supplementary material, making it hard to identify the quality of the generated videos.

6. Minor (non-exhaustive): many typos, e.g. L87, L93, L95 ... Figure 4(a) ..., L176, inproves,etc.

**Questions:**

Please see the weaknesses.

---

### Official Review · Reviewer_PNbY · 2025-10-31

**Soundness:** 3
**Presentation:** 2
**Contribution:** 2
**Rating:** 4
**Confidence:** 3

**Summary:**

In this paper, authors propose Ent-Prog, an efficient training framework for diffusion models for pose-guided human video generation. Ent-Prog aims to reduce computational costs with CEI, which prioritizes the training of critical network blocks, with an adaptive progressive schedule that adjusts the number of unfrozen blocks based on convergence efficiency. Experiments across three human video datasets show that Ent-Prog achieves up to a 2.2× training speedup and a 2.4× GPU memory reduction while maintaining or enhancing generative performance compared to full training.

**Strengths:**

1. The proposed method delivers impressive practical benefits, achieving up to a 2.2× training speedup and a 2.4× GPU memory reduction.
2. CEI is a new metric for assessing the importance of blocks.
3. The introduction of the adaptive progressive schedule using a Nested Diffusion Supernet to dynamically select the optimal number of blocks to unfreeze based on convergence efficiency is well-motivated and demonstrates superior performance compared to simply removing the schedule.

**Weaknesses:**

1. Missing CEI Distribution Analysis: The paper lacks an in-depth analysis or visualization of CEI distribution to confirm whether only a few blocks truly have high priority scores, which is crucial.
2. Missing details of baseline
3. Missing comparison to LoRA-based finetuning and other related works.
4. Missing the comparison on training time (hours)
5. How is the proposed method special for human video generation? This seems to be a general training accelration method to me.
6. How does the proposed method generalize among different baseline methods? Authors should test the model on different base models.

**Questions:**

1. Line 093/95/87: Is figure 4 actually figure 2?
2. Since in FIgure 2 (a/b/c), blocks are randomly selected, there should be variance on the results right?
3. See weakness.

---

### Official Review · Reviewer_CXVX · 2025-11-01

**Soundness:** 3
**Presentation:** 3
**Contribution:** 3
**Rating:** 4
**Confidence:** 3

**Summary:**

The paper develops an efficient fine-tuning method for the specific task of human video. It introduces CEI-based prioritized progressive learning, which quantifies the training priorities of different blocks, enabling efficient and high-quality fine-tuning.

**Strengths:**

- Leveraging CEI for PPL demonstrates a certain level of insight and novelty.
- The experimental results show a significant improvement, balancing both quality and training efficiency.
- The paper is well-organized and easy to read.

**Weaknesses:**

I am concerned that there may be a critical factual error in the paper. The authors claim in line 206,207 that `Since diffusion models are trained to predict Gaussian noise, we expect the distribution of ε to be approximately Gaussian`, which is the key to applying CEI in practice. However, the epsilon-prediction in diffusion models does not follow a Gaussian distribution. On the contrary, it is highly related to the data distribution [1]. Given this, how can the authors use Eq. (2) to approximate the true CEI?

This is the main reason why I gave a rating of 4. If the authors can provide a theoretical clarification, I'm open to raising the rating.

[1] Score-Based Generative Modeling through Stochastic Differential Equations
## Questions
CEI should be applicable to any conditional model, so the question arises whether the method presented in this paper can be generalized to other controllable generation tasks, such as camera-pose [2]. If the paper can symbolically demonstrate the method's generalizability on a toy dataset, it would significantly enhance the contribution of the work.

[2] CameraCtrl: Enabling Camera Control for Video Diffusion Models

**Questions:**

See Weakness

---

### Official Review · Reviewer_ZVpD · 2025-11-03

**Soundness:** 1
**Presentation:** 1
**Contribution:** 2
**Rating:** 2
**Confidence:** 3

**Summary:**

This paper addresses the high computational cost of training human video generation and proposes Entropy-Guided Prioritized Progressive Learning (Ent-Prog), which estimates block importance via Conditional Entropy Inflation (CEI) to progressively unfreeze high-priority blocks and uses a Nested Diffusion Supernet to adapt the unfreezing width based on convergence efficiency. Experiments on human-video benchmarks show faster training and lower GPU memory while maintaining or improving visual quality and pose adherence.

**Strengths:**

- The idea of estimating block importance and progressively unfreezing from high-priority blocks is intuitive and straightforward, and its ease of being retrofitted to existing models makes it practically useful.
- This adaptive scheduling reduces trial-and-error, balances quality and compute under a fixed budget, and is validated by experiments, highlighting its practical utility.

**Weaknesses:**

- While the paper focuses on human video generation, the proposed approach, estimating block importance and progressively unfreezing high-priority components, appears domain-agnostic. The authors should clarify what adaptations, if any, are specific to the human-video setting (e.g., choices tied to identity preservation or pose adherence). Alternatively, if the method is intended to be general, its effectiveness should be validated on additional video generation tasks beyond humans.
- The paper reports internal baselines and ablations, but it does not compare against representative parameter-efficient tuning methods (e.g., LoRA/Adapters) or other progressive/elastic training schemes. Without side-by-side comparisons on both model performance and total training cost against these, it is difficult to conclude that the proposed method is preferable in practice. In addition, the paper lacks a clear section that situates the work relative to existing literature and articulates its advantages over prior methods.
- The paper uses CEI to rank blocks by training priority, yet it does not provide a principled or empirical justification for why CEI, specifically, should be preferred. Figure 2 shows a correlation between CEI and loss changes when freezing blocks during training, but correlation under a shared intervention does not establish that CEI is uniquely appropriate; similar phenomena could arise for other reasonable metrics (e.g., gradient norm/energy, Fisher information, Hessian approximations, mutual information with conditioning signals, or SNR-style measures). To avoid circular reasoning, the authors should clarify the theoretical basis for adopting CEI over other metrics.
- Several presentation issues hinder readability and reproducibility (assuming these observations are accurate):
  - The introduction refers to “Figure 4 (a),” but Figure 4 contains no panel (a).
  - Figure 1 is never cited in the text.
  - CEI is mentioned in the paragraph before the paper states “we introduce CEI,” which breaks the flow of the introduction.

**Questions:**

Regarding the weaknesses noted above, please respond and include additional experiments to address them.

---

### Note · Authors · 2025-11-14

I have read and agree with the venue's withdrawal policy on behalf of myself and my co-authors.